# Judd-Ofelt Analysis of High Erbium Content Yttrium-Aluminum and Yttrium-Scandium-Aluminum Garnet Ceramics

Vadim Zhmykhov [1], Elena Dobretsova [1,*], Vladimir S. Tsvetkov [1], Marina Nikova [2], Irina Chikulina [2], Dmitry Vakalov [2], Vitaly Tarala [2], Yurii Pyrkov [1], Sergey Kuznetsov [1,3] and Vladimir Tsvetkov [1]

1 Prokhorov General Physics Institute of the Russian Academy of Sciences, 119991 Moscow, Russia
2 Scientific and Laboratory Complex Clean Room, North Caucasus Federal University, 355029 Stavropol, Russia
3 Kazan Federal University, 420008 Kazan, Russia
* Correspondence: elenadobretsova89@gmail.com or eadobr@kapella.gpi.ru

**Abstract:** The $Er_{1.5}Y_{1.5}Al_5O_{12}$ (Er:YAG) and $(Er_{1.43}Y_{1.43}Sc_{0.14})(Sc_{0.24}Al_{1.76})Al_3O_{12}$ (Er:YSAG) ceramics have been characterized using the Judd-Ofelt (JO) theory. The line strengths and oscillator strengths of several transitions from the ground state $^4I_{15/2}$ to excited state manifolds have been evaluated from transmittance spectra measured at room temperature (300 K). The JO parameters have been calculated, and the values of the radiative decays rate and the radiative lifetimes for the $^4I_{13/2}$ excited state, and the luminescence cross-section of $^4I_{15/2} \rightarrow ^4I_{13/2}$ in Er-doped ceramic samples have been established. We have traced the influence of $Sc^{3+}$ inclusion on spectroscopic properties and crystal quality and estimate prospects of application in laser systems.

**Keywords:** yttrium-scandate-aluminum garnet; transparent ceramics; Judd-Ofelt parameters

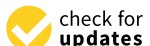



## 1. Introduction

Rare-earth-doped yttrium-aluminum garnet $Y_3Al_5O_{12}$ materials (YAG) are well-known as active media for solid-state lasers [1–4]. In recent years, rare-earth-doped materials have been broadly used in solid-state lasers, colour displays [5–7], optical amplifiers [8], free-space optical communications [9,10], sensors [11–13], 3D waveguides [14,15], and X-ray screens [16]. Recently, $Er^{3+}$:YAG has attracted considerable interest for its high-power, high-energy eye-safe lasers operating near 1.5 μm for range finding, flash lidar, and other remote-sensing applications [17]. Emitting at mid-infrared 2.9 μm Er:YAG has found wide application in medicine—in gynecology [18,19], dentistry [20], and microsurgery [21].

Transparent ceramics were initially developed to replace single crystals in cases of disk geometry and multilayer and concentration gradient architectures. Among solid-state lasers, disk lasers offer significant advantages for both ultrafast and continuous wave operation [22]. In a disk laser, the gain medium is shaped like a disk with a large diameter in comparison to its thickness. This geometry allows the gain medium to be very efficiently cooled. The large mode areas on the gain medium and the short propagation distance of the pulses through the gain medium make it inherently advantageous for small nonlinearities at very high pulse energies. Mode-locked thin-disk oscillators have consistently achieved orders of magnitude higher than the average power and pulse energy of other narrow pulse and ultranarrow pulse oscillator technology, reaching comparable levels to advanced high-power amplifiers operating at an MHz repetition rate [23].

The doping concentration of rare-earth ions in YAG ceramics can reach 100% [24,25]. A promising feature of YSAG ceramics is the introduction of scandium into the dodecahedral and octahedral positions of the garnet that leads to disordering of the crystal lattice. This is expressed in the broadening of the absorption bands that may results in achievement of laser pulses with high power and short duration. These lasing parameters are useful because they decrease medical laser interventions and are therefore less traumatic.

In this paper, we report optical characterization of $Er_{1.5}Y_{1.5}Al_5O_{12}$ and $(Er_{1.43}Y_{1.43}Sc_{0.14})$ $(Sc_{0.24}Al_{1.76})Al_3O_{12}$ ceramics carried out through a comparative Judd-Ofelt (JO) analysis [26,27]. The Judd-Ofelt parameters were calculated from the experimental absorption spectra of the ceramics to trace the influence of $Sc^{3+}$ inclusion on spectroscopic properties and crystal quality and estimate the prospects for application in laser systems.

## 2. Results and Discussion

The normalized transmittance spectrum and absorption coefficient for $Er^{3+}$:YSAG ceramic samples ($S_4$) are given in Figure 1. The spectra consist of eight complex $Er^{3+}$ ($^4f_{11}$) lines, which are observed at 372.9, 407.8, 448.7, 513.7, 653.2, 798.5, 968.1, and 1494.1 nm and correspond to $^4I_{15/2} \rightarrow {}^4G_{11/2} + {}^4G_{9/2} + {}^2K_{15/2} + {}^2G_{7/2}, {}^2H_{9/2}, {}^4F_{5/2} + {}^4F_{3/2}, {}^4S_{3/2} + {}^2H_{11/2}$ $+ {}^4F_{7/2}, {}^4F_{9/2}, {}^4I_{9/2}, {}^4I_{11/2},$ and $^4I_{13/2}$ electron transitions, respectively.

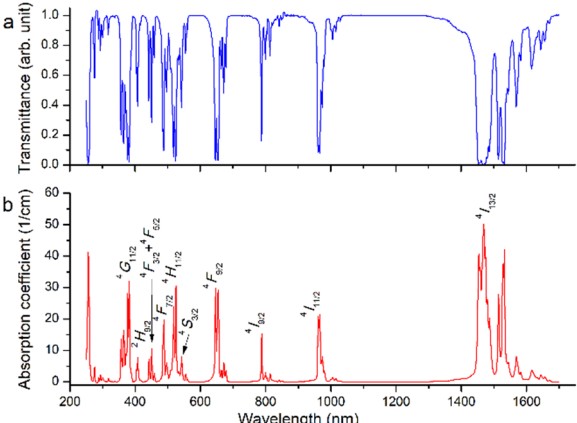

**Figure 1.** (**a**) Normalized transmittance spectrum and (**b**) absorption coefficient of the $Er^{3+}$: YSAG ceramics ($S_4$).

The room-temperature absorption spectra of $Er^{3+}$-doped YAG/YSAG ceramics in the spectral range of 770–860 nm corresponding to $^4I_{15/2} \rightarrow {}^4I_{9/2}$ electron transitions in $Er^{3+}$ ion are presented in Figure 2. The broadening of spectral lines is recognized as result of the partially disordering crystal structure of the ceramics due to $Sc^{3+}$ doping.

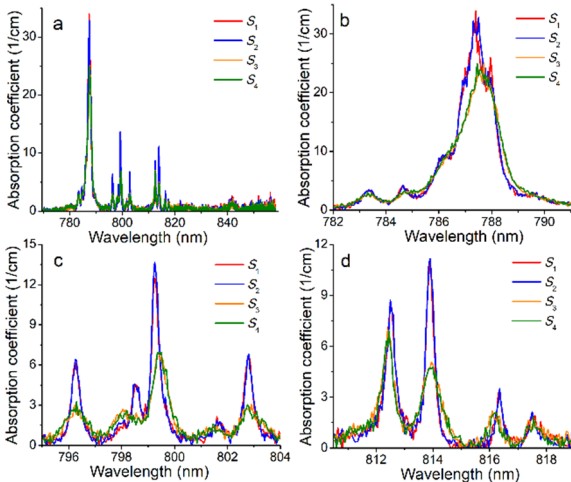

**Figure 2.** Room-temperature absorption spectra of Er:YAG ($S_1$, $S_2$) and Er:YSAG ($S_3$, $S_4$) ceramics in the range of $^4I_{15/2} \rightarrow {}^4I_{9/2}$ electron transitions: (**a**) 770–860 nm, (**b**) 780–900 nm, (**c**) 795–805 nm, and (**d**) 810–820 nm.

JO analysis has been applied to estimate the spectroscopic parameter changes in Er:YSAG ceramics in comparison with Er:YAG ones. Eight complex lines of the room-

temperature transmittance spectra were chosen to determine the JO parameters for the corresponding $Er^{3+}$ ($^4f_{11}$) transitions in samples.

The mean wavelengths and integrated absorption coefficients of Er:YAG and Er:YSAG ceramics (Table 1) were used for the determining of line strengths $s_{meas}$ and oscillator strengths $f_{meas}$ (Table 2). The determined line strengths are used to calculate $\Omega_2$, $\Omega_4$, and $\Omega_6$ parameters by the Judd-Ofelt theory. The values of the measured ($s_{meas}$) and calculated ($s_{calc}$) absorption line strengths are tabulated in Table 2. The values of the measured ($f_{meas}$) and calculated ($f_{calc}$) absorption oscillator strengths are summarized in Table 3. The values of root mean square (RMS) and relative error for the oscillator strength and absorption oscillator strengths are listed in Tables 2 and 3, respectively. The values of JO parameters are given in Table 4. The spectral intensity parameters are reflected in many crystal effects such as chemical bonds between the ions in the host, the charge distribution on the ions in the cell, and the lattice distortion. The $\Omega_2$ value is the most affected to changes in the environment of the lanthanide ion. The $\Omega_2$ value is higher in Er:YSAG compared with Er:YAG due to substitution of $Y^{3+}$ by $Sc^{3+}$ ions and local distortion in the dodecahedral position. Meanwhile, the other two JO parameters of Er:YSAG ceramics are similar to corresponding parameters of the Er:YAG ones and vary insignificantly. This behavior can be explained by the fact that $\Omega_6$ and $\Omega_4$ are usually more sensitive to change in $f$-electron number (changing a type of a rare-earth ion) and are less affected (or unaffected) by the environment [28].

**Table 1.** The mean wavelengths ($\overline{\lambda}$) and integrated absorption coefficient ($\Gamma$) corresponding to electron transitions of $Er^{3+}$:YAG ($Er^{3+}$:YSAG) ceramic samples at 300 K.

| Transition $^4I_{15/2} \rightarrow$ | $S_1$ | | $S_2$ | | $S_3$ | | $S_4$ | |
|---|---|---|---|---|---|---|---|---|
| | $\overline{\lambda}$ (nm) | $\Gamma$ (nm × cm$^{-1}$) | $\overline{\lambda}$ (nm) | $\Gamma$ (nm × cm$^{-1}$) | $\overline{\lambda}$ (nm) | $\Gamma$ (nm × cm$^{-1}$) | $\overline{\lambda}$ (nm) | $\Gamma$ (nm × cm$^{-1}$) |
| $^4I_{13/2}$ | 1495.0 | - | 1494.9 | - | 1494.1 | - | 1494.1 | - |
| $^4I_{11/2}$ | 968.8 | 268.250 | 967.2 | 275.135 | 968.4 | 295.429 | 968.1 | 297.084 |
| $^4I_{9/2}$ | 797.5 | 100.318 | 796.1 | 90.996 | 795.9 | 97.574 | 798.5 | 110.707 |
| $^4F_{9/2}$ | 653.5 | 339.720 | 653.3 | 331.758 | 653.2 | 379.385 | 653.2 | 376.760 |
| $^4S_{3/2}+^2H_{11/2}+^4F_{7/2}$ | 514.1 | 491.496 | 514.7 | 456.811 | 514.1 | 489.456 | 513.7 | 496.346 |
| $^4F_{5/2}+^4F_{3/2}$ | 449.9 | 78.839 | 448.6 | 71.185 | 448.5 | 76.869 | 448.7 | 78.585 |
| $^2H_{9/2}$ | 408.5 | 44.988 | 407.7 | 41.845 | 407.8 | 45.055 | 407.8 | 43.392 |
| $^4G_{11/2}+^4G_{9/2}+^2K_{15/2}+^2G_{7/2}$ | 372.8 | 389.977 | 372.6 | 385.820 | 372.8 | 383.785 | 372.9 | 374.759 |

**Table 2.** Values of the measured and calculated absorption line strengths of $Er^{3+}$ in the YAG (YSAG) ceramic samples at 300 K; *ed* is electric dipole transition, *md* is magnetic dipole one.

| Transition $^4I_{15/2} \rightarrow$ | $S_1$ | | $S_2$ | | $S_3$ | | $S_4$ | |
|---|---|---|---|---|---|---|---|---|
| | $s_{exp} \times 10^{-20}$ (cm$^2$) | $s_{calc} \times 10^{-20}$ (cm$^2$) | $s_{exp} \times 10^{-20}$ (cm$^2$) | $s_{calc} \times 10^{-20}$ (cm$^2$) | $s_{exp} \times 10^{-20}$ (cm$^2$) | $s_{calc} \times 10^{-20}$ (cm$^2$) | $s_{exp} \times 10^{-20}$ (cm$^2$) | $s_{calc} \times 10^{-20}$ (cm$^2$) |
| $^4I_{13/2}$ | - | 1.97(16) $^{ed}$ +0.72 $^{md}$ | - | 1.94(17) $^{ed}$ +0.72 $^{md}$ | - | 2.19(20) $^{ed}$ +0.72 $^{md}$ | - | 2.34(11) $^{ed}$ +0.72 $^{md}$ |
| $^4I_{11/2}$ | 0.38 | 0.26(2) | 0.42 | 0.27(3) | 0.42 | 0.25(2) | 0.40 | 0.28(1) |
| $^4I_{9/2}$ | 0.19 | 0.14(5) | 0.17 | 0.16(1) | 0.17 | 0.15(1) | 0.18 | 0.15(1) |
| $^4F_{9/2}$ | 0.71 | 0.72(9) | 0.79 | 0.79(7) | 0.73 | 0.73(7) | 0.75 | 0.76(4) |
| $^4S_{3/2}+^2H_{11/2} + ^4F_{7/2}$ | 1.19 | 1.26(1) | 1.29 | 1.36(1) | 1.27 | 1.36(1) | 1.14 | 1.43(7) |
| $^4F_{5/2} + ^4F_{3/2}$ | 0.25 | 0.22(3) | 0.23 | 0.23(2) | 0.22 | 0.21(2) | 0.25 | 0.23(1) |
| $^2H_{9/2}$ | 0.13 | 0.16(1) | 0.15 | 0.17(2) | 0.14 | 0.15(1) | 0.15 | 0.16(1) |
| $^4G_{11/2}+^4G_{9/2} + ^2K_{15/2} + ^2G_{7/2}$ | 1.24 | 1.19(4) | 1.35 | 1.292(1) | 1.43 | 1.35(1) | 1.44 | 1.39(8) |
| RMS $\Delta s$ | $5.97 \times 10^{-2}$ | | $7.13 \times 10^{-2}$ | | $7.32 \times 10^{-2}$ | | $8.25 \times 10^{-2}$ | |
| RMS error (%) | 8.17 | | 9.02 | | 9.19 | | 5.05 | |

**Table 3.** Values of the measured and calculated absorption oscillator strengths of $Er^{3+}$ in the YAG (YSAG) ceramic samples at 300 K; *ed* is electric dipole transition, *md* is magnetic dipole one.

| Transition $^4I_{15/2}\rightarrow$ | $S_1$ | | $S_2$ | | $S_3$ | | $S_4$ | |
|---|---|---|---|---|---|---|---|---|
| | $f_{exp} \times 10^{-6}$ | $f_{calc} \times 10^{-6}$ | $f_{exp} \times 10^{-6}$ | $f_{calc} \times 10^{-6}$ | $f_{exp} \times 10^{-6}$ | $f_{calc} \times 10^{-6}$ | $f_{exp} \times 10^{-6}$ | $f_{calc} \times 10^{-6}$ |
| $^4I_{13/2}$ | - | 1.54(20) $^{ed}$ +0.59 $^{md}$ | - | 1.51(5) $^{ed}$ +0.59 $^{md}$ | - | 1.71(7) $^{ed}$ +0.59 $^{md}$ | - | 1.84(11) $^{ed}$ +0.59 $^{md}$ |
| $^4I_{11/2}$ | 0.46 | 0.31(4) | 0.52 | 0.34(1) | 0.50 | 0.30(1) | 0.49 | 0.33(7) |
| $^4I_{9/2}$ | 0.26 | 0.17(2) | 0.25 | 0.24(1) | 0.25 | 0.22(1) | 0.27 | 0.22(5) |
| $^4F_{9/2}$ | 1.28 | 1.32(2) | 1.46 | 1.45(5) | 1.33 | 1.33(6) | 1.36 | 1.37(4) |
| $^4S_{3/2} + {}^2H_{11/2} + {}^4F_{7/2}$ | 3.00 | 2.97(4) | 3.03 | 3.20(2) | 2.95 | 3.18(2) | 3.18 | 3.35(2) |
| $^4F_{5/2} + {}^4F_{3/2}$ | 0.63 | 0.61(8) | 0.63 | 0.65(1) | 0.61 | 0.57(3) | 0.67 | 0.63(2) |
| $^2H_{9/2}$ | 0.44 | 0.48(7) | 0.44 | 0.51(2) | 0.43 | 0.46(2) | 0.46 | 0.50(2) |
| $^4G_{11/2} + {}^4G_{9/2} + {}^2K_{15/2} + {}^2G_{7/2}$ | 4.53 | 3.96(7) | 4.52 | 4.34(2) | 4.76 | 4.51(2) | 4.80 | 4.63(2) |
| RMS $\Delta f$ | $2.920 \times 10^{-1}$ | | $0.785 \times 10^{-1}$ | | $1.004 \times 10^{-1}$ | | $0.769 \times 10^{-1}$ | |
| RMS error (%) | 13.6 | | 3.64 | | 4.54 | | 3.39 | |

**Table 4.** Judd–Ofelt parameters of the $Er^{3+}$-doped YAG (YSAG) ceramics at 300 K.

| Ceramic Samples | Judd-Ofelt Parameters | | |
|---|---|---|---|
| | $\Omega_2 \times 10^{-20}$, $cm^2$ | $\Omega_4 \times 10^{-20}$, $cm^2$ | $\Omega_6 \times 10^{-20}$, $cm^2$ |
| $S_1$ | 0.2955 | 0.8037 | 0.6410 |
| $S_2$ | 0.3064 | 0.8984 | 0.6818 |
| $S_3$ | 0.4446 | 0.8521 | 0.6080 |
| $S_4$ | 0.4681 | 0.8378 | 0.6741 |

Room-temperature luminescence spectra of all ceramic samples in comparison with the $Er^{3+}$:YAG single crystal are shown in Figure 3. The line widths of the YAG samples $S_1$ and $S_2$ are comparable with the single crystal one. Line widths of the ceramics containing $Sc^{3+}$ ($S_3$, $S_4$) look broader due to the higher degree of disorder in the crystal structure. We observe a shift of the line maxima in Er:YSAG ceramic samples relative to Er:YAG ceramics and the single crystal.

The radiative decay rate ($A_{rad}$) and the radiative decay time ($\tau_{rad}$) for the $^4I_{13/2} \rightarrow {}^4I_{15/2}$ electron transition has been evaluated using the JO parameters (Table 5). To obtain more reliable information about the perspective of application of our materials, we have calculated the values of an emission cross-section of the $^4I_{13/2} \rightarrow {}^4I_{15/2}$ electron transition.

**Table 5.** Radiative decay rates ($A_{J \rightarrow J'}$), radiative decay time $\tau_{rad}^{calc}$, fluorescence lifetime ($\tau_{lum}^{exp}$), the intrinsic quantum yield ($\eta$), non-radiative multiphonon decay rates ($W_{NR}$) of the $^4I_{13/2}$ multiplet, and the values of emission cross section ($\sigma$) of the $^4I_{13/2} \rightarrow {}^4I_{15/2}$ electron transition in $Er^{3+}$-doped YAG (YSAG) ceramics at 300 K.

| Parameters | $S_1$ | $S_2$ | $S_3$ | $S_4$ |
|---|---|---|---|---|
| $A_{J \rightarrow J'}$ ($s^{-1}$) | 137.74 | 143.98 | 133.08 | 140.26 |
| $\tau_{rad}^{calc}$ (ms) | 3.2 | 3.1 | 3.2 | 3.1 |
| $\tau_{lum}^{exp}$ (ms) | 0.146 (3) | 0.180 (2) | 0.264 (4) | 0.266 (3) |
| $\eta$ (%) | 4.56 | 5.81 | 8.25 | 8.58 |
| $W_{NR}$ ($s^{-1}$) | 6536 | 5233 | 3475 | 3437 |
| $\sigma \cdot 10^{-19}$ ($cm^2$) | 0.105 | 0.110 | 0.097 | 0.082 |

Another important parameter is the intrinsic quantum yield ($\eta$) being evaluated from the ratio of the fluorescence to radiative decay time (Formula (S13), Supporting Information). Figure 4 shows decay curves being measured for the $^4I_{13/2} \rightarrow {}^4I_{15/2}$ electron transition in Er:YAG and Er:YSAG ceramics. The fluorescence lifetimes and the intrinsic quantum yield being obtained for the $^4I_{13/2} \rightarrow {}^4I_{15/2}$ electron transition are given in Table 5.

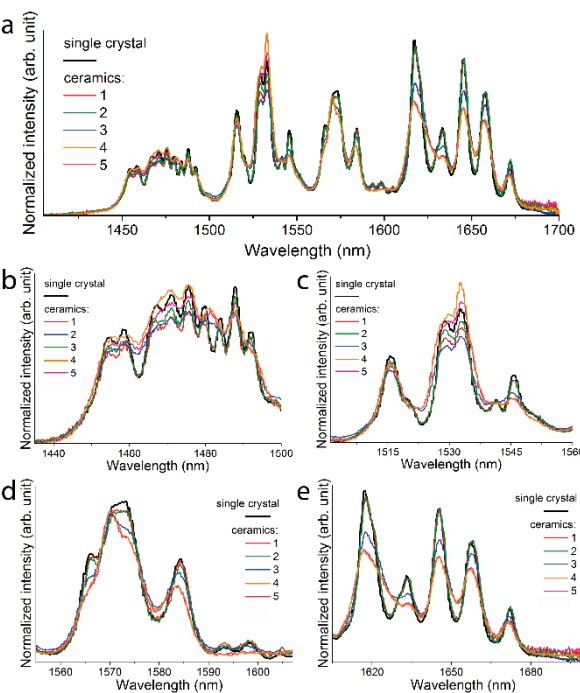

**Figure 3.** Room-temperature luminescence spectra of Er:YAG single crystal, Er:YAG ($S_1$, $S_2$) and Er:YSAG ($S_3$, $S_4$) ceramics measured in the range of: (**a**) 1400–1700 nm ($^4I_{13/2} \rightarrow {}^4I_{15/2}$ transition), (**b**) 1440–1500 nm, (**c**) 1500–1560 nm, (**d**) 1560–1610 nm, and (**e**) 1605–1690 nm.

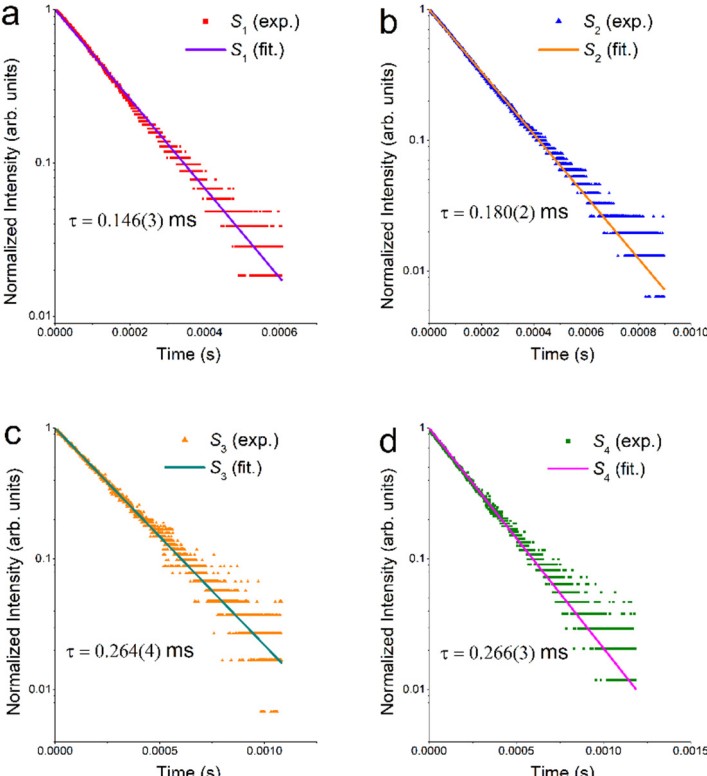

**Figure 4.** Decay curves being obtained experimentally for $^4I_{13/2} \rightarrow {}^4I_{15/2}$ electron transition in Er:YAG ($S_1$, $S_2$) ((**a**,**b**), respectively) and Er:YSAG ($S_3$, $S_4$) ((**c**,**d**), respectively) ceramics at 300 K.

The non-radiative multiphonon decay rates may be given by Expression (S14) (Supporting Information). The main contribution to non-radiative decay for the ceramic samples with high dopant comes from multiphonon relaxation from the host and energy

transfer interaction between nearby ions. Moreover, numerous grain boundaries can act as quenchers.

### 3. Materials and Methods

The ceramic samples have been synthesized by co-precipitation from an aqueous solution following annealing at a high temperature according to earlier published protocol [29–31]. We investigated four ceramic samples of the following chemical composition: $Er_{1.5}Y_{1.5}Al_5O_{12}$ ($S_1$ and $S_2$ samples) and $Er_{1.43}Y_{1.43}Sc_{0.38}Al_{4.76}O_{12}$ ($S_3$ and $S_4$ samples). Values of uniaxial pressing vary in the range of 50–100 MPa, and temperature of vacuum sintering—in the range of 1760–1780 °C [32]. The Er:YAG and Er:YSAG ceramics looked like disks with a diameter of 10 mm and thickness of about 1 mm.

The room-temperature transmittance spectra of the $Er^{3+}$:YAG ceramics were recorded using the Shimadzu UW-3101PC spectrophotometer controlled by a desktop computer in the range of 250–1700 nm with resolution of 1 nm. The high-resolved (0.1 nm) transmittance spectra have been measured in the range of 770 to 860 nm ($^4I_{15/2} \rightarrow {}^4F_{9/2}$ electron transitions in $Er^{3+}$ ion).

Transmittance spectra of the ceramics have been analyzed using JO theory. The analysis is described in detail in Supporting Information. The absorption coefficients of the samples are calculated by Equation (1) using experimental transition spectra:

$$\alpha(\lambda) = (I/I_0) \tag{1}$$

where $\alpha(\lambda)$ is the absorption coefficient; $I$ and $I_0$ are spectral intensities of the light transmitted through the sample and directed into the sample, respectively; $l$ is the thickness of the sample.

The room-temperature fluorescence spectra have been recorded in the range of 1400 to 1700 nm corresponding to $^4I_{13/2} \rightarrow {}^4I_{15/2}$ electron transitions in $Er^{3+}$:YAG/YSAG ceramics. The $Er^{3+}$:YAG single crystal has been grown at the Research Institute of Materials Science and Technology (Zelenograd, Russia) and used as a standard. The spectra have been measured using the ARC SpectraPro-300i monochromator at diode laser excitation at a wavelength of 965 nm and irradiation of up to 3 mW. The signals were detected with a thermoelectrically cooled InGaAs detector.

### 4. Conclusions

A spectroscopic analysis of $Er^{3+}$ in YAG ($S_1$, $S_2$) and YSAG ($S_3$, $S_4$) ceramics has been performed by the Judd-Ofelt theory. The Judd-Ofelt parameters such as $\Omega_2$, $\Omega_4$, and $\Omega_6$ are determined for Er:YAG ($S_1$, $S_2$) and Er:YSAG ($S_3$, $S_4$) ceramics. The $\Omega_2$ value is increased in $S_1$ to $S_4$ samples due to asymmetry of the crystal field around $Er^{3+}$ and increases with a disorder degree in the crystal structure. The predicted radiative decay time of the $^4I_{13/2}$ electron level varies insignificantly. Disordering in YSAG ceramics results in the broadening the emission lines and smoothing of the luminescence/gain spectrum in the 1.5-μm range in comparison with the YAG crystal or ceramic host. This factor makes it a promising active media for the amplifiers in this spectral range. The shorter experimental lifetimes and relatively low intrinsic quantum yield in the heavily doped $Er^{3+}$:YSAG ceramics can be attributed to the concentration effect, where the energy up-conversion and cross-relaxation mechanisms become increasingly important. However, the heavily doped $Er^{3+}$:YSAG ceramics can therefore be considered as an excellent active media for a 3-μm laser system.

**Supplementary Materials:** The following supporting information can be downloaded at: https://www.mdpi.com/article/10.3390/inorganics10100170/s1, Supporting Information include basic expressions being used for the Judd-Ofelt analysis of the transition spectra. References [26–28,33–36] are cited in the Supplementary Materials.

**Author Contributions:** Conceptualization, V.T. (Vitaly Tarala), E.D., S.K. and V.T. (Vladimir Tsvetkov); data curation, V.Z. and E.D.; formal analysis, V.Z. and V.S.T.; funding acquisition, E.D. and S.K.; investigation, V.Z., E.D. and Y.P., methodology, V.T. (Vitaly Tarala) and V.T. (Vladimir Tsvetkov);

project administration, E.D.; resources, M.N., I.C., D.V. and V.T. (Vitaly Tarala); software, V.Z. and E.D.; supervision, V.T. (Vladimir Tsvetkov); validation, V.Z. and E.D.; visualization, V.Z. and E.D.; writing—original draft preparation, V.Z. and E.D.; writing—review and editing, E.D., S.K. and V.T. (Vladimir Tsvetkov). All authors have read and agreed to the published version of the manuscript.

**Funding:** The work on ceramics fabrication is carried out in accordance with the Strategic Academic Leadership Program "Priority 2030" of the Kazan Federal University of the Government of the Russian Federation. The spectral measurements and the Judd-Ofelt analysis were supported by the Grant of the President of the Russian Federation NoK-72.2022.1.2.

**Data Availability Statement:** Not applicable.

**Conflicts of Interest:** The authors declare no conflict of interest. The funders had no role in the design of the study; in the collection, analyses, or interpretation of data; in the writing of the manuscript; or in the decision to publish the results.

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
