# Peer review of "Judd-Ofelt Analysis of High Erbium Content Yttrium-Aluminum and Yttrium-Scandium-Aluminum Garnet Ceramics"

_inorganics, doi:10.3390/inorganics10100170_

Round 1

Reviewer 1 Report

I have read with interest this nice paper concerning photonic ceramics. The paper sounds well. The experiments are clearly presented and the results are discussed with competence. Cross section and other parameters are of certain interest. My concern is related to lifetime values. Using JO the lifetimes of the different systems are assessed. However, I do not find decay curves in the text. I think that comparison between experiment and calculation could enrich the paper.

Author Response

Response to Reviewer 1 Comments

I have read with interest this nice paper concerning photonic ceramics. The paper sounds well. The experiments are clearly presented and the results are discussed with competence. Cross section and other parameters are of certain interest. My concern is related to lifetime values. Using JO the lifetimes of the different systems are assessed. However, I do not find decay curves in the text. I think that comparison between experiment and calculation could enrich the paper.

Response: Thank the reviewer for his kind comments. In the updated version of the manuscript, we have added decay curves and the values being obtained experimentally for lifetimes. The shorter experimental lifetimes in the heavily doped Er3+:YAG and Er3+:YSAG ceramics can be attributed to the concentration effect, where the energy up-conversion and cross-relaxation mechanisms become increasingly important. However, the heavily Er3+-doped ceramics can therefore be considered as an excellent active media for a 3-µm laser system.

Reviewer 2 Report

The paper titled: Judd-Ofelt analysis of high erbium content yttrium-aluminum and yttrium-scandium-aluminum garnet ceramics“ presents excellent research conducted on a very important material with high applicability. I recommend publication after minor revision. My comments to the authors are given below.

There seems to be an error with the spectroscopic quality factor being propagated in works other than those concerned with Nd3+. To quote Warland and Binnemans in 10.1016/S0168-1273(98)25006-9: “For other lanthanide ions than Nd 3+, the spectroscopic quality factor is not usable, since in general the branching ratio cannot be reduced to a function of the f24/f26 ratio.”. Thus, I suggest dropping the spectroscopic quality factor completely.

Another important parameter is missing, and that is the intrinsic quantum yield. The explanation for it is given, for example, in Eq. 6 in 10.1016/j.jlumin.2018.09.048 or Eq. 28 in 10.1016/j.ccr.2015.02.015 (those papers are for Eu3+, but the eq. for intrinsic quantum yield is universal for all lanthanides). This would mean that the lifetimes would have to be measured. For this kind of research paper it is highly desirable, as many quantities, like nonradiative rate (see the same papers), can be calculated. All those parameters are important for laser systems, which is the main application of this material.

Author Response

Response to Reviewer 2 Comments

The paper titled: “Judd-Ofelt analysis of high erbium content yttrium-aluminum and yttrium-scandium-aluminum garnet ceramics“ presents excellent research conducted on a very important material with high applicability. I recommend publication after minor revision. My comments to the authors are given below.

There seems to be an error with the spectroscopic quality factor being propagated in works other than those concerned with Nd3+. To quote Warland and Binnemans in 10.1016/S0168-1273(98)25006-9: “For other lanthanide ions than Nd 3+, the spectroscopic quality factor is not usable, since in general the branching ratio cannot be reduced to a function of the f24/f26 ratio.”. Thus, I suggest dropping the spectroscopic quality factor completely.

Another important parameter is missing, and that is the intrinsic quantum yield. The explanation for it is given, for example, in Eq. 6 in 10.1016/j.jlumin.2018.09.048 or Eq. 28 in 10.1016/j.ccr.2015.02.015 (those papers are for Eu3+, but the eq. for intrinsic quantum yield is universal for all lanthanides). This would mean that the lifetimes would have to be measured. For this kind of research paper it is highly desirable, as many quantities, like nonradiative rate (see the same papers), can be calculated. All those parameters are important for laser systems, which is the main application of this material.

Response: Thank the reviewer for his kind comments. We have updated the manuscript and added values of experimentally obtained lifetimes, the intrinsic quantum yield and the non-radiative multiphonon decay rates. We have deleted the spectroscopic quality factor completely. The shorter experimental lifetimes and relatively low intrinsic quantum yield in the heavily doped Er3+:YAG ceramics can be attributed to the concentration effect, where the energy up-conversion and cross-relaxation mechanisms become increasingly important. However, the heavily doped Er3+:YSAG ceramics can therefore be considered as an excellent active media for a 3-µm laser system.